# Understanding the Role of LFA-1 in Leukocyte Adhesion Deficiency Type I (LAD I): Moving towards Inflammation?

**DOI:** 10.3390/ijms23073578

**Published:** 2022-03-25

**Authors:** Julia Fekadu, Ute Modlich, Peter Bader, Shahrzad Bakhtiar

**Affiliations:** 1Department for Children and Adolescents, Division for Stem Cell Transplantation, Immunology and Intensive Care Medicine, Goethe University, 60590 Frankfurt, Germany; julia.fekadu@kgu.de (J.F.); peter.bader@kgu.de (P.B.); 2Research Group for Gene Modification in Stem Cells, Division of Veterinary Medicine, Paul-Ehrlich-Institute, 63225 Langen, Germany; ute.modlich@pei.de

**Keywords:** inborn errors of immunity, inflammation, hematopoietic stem cell transplantation

## Abstract

LFA-1 (Lymphocyte function-associated antigen-1) is a heterodimeric integrin (CD11a/CD18) present on the surface of all leukocytes; it is essential for leukocyte recruitment to the site of tissue inflammation, but also for other immunological processes such as T cell activation and formation of the immunological synapse. Absent or dysfunctional expression of LFA-1, caused by mutations in the *ITGB2* (integrin subunit beta 2) gene, results in a rare immunodeficiency syndrome known as Leukocyte adhesion deficiency type I (LAD I). Patients suffering from severe LAD I present with recurrent infections of the skin and mucosa, as well as inflammatory symptoms complicating the clinical course of the disease before and after allogeneic hematopoietic stem cell transplantation (alloHSCT); alloHSCT is currently the only established curative treatment option. With this review, we aim to provide an overview of the intrinsic role of inflammation in LAD I.

## 1. LFA-1 and Primary Immunodeficiency Syndromes 

Leukocyte adhesion and migration to the site of inflammation involve a well-defined interplay between different selectins, chemokines, and integrins [1]. Up to now, four primary immunodeficiency syndromes affecting the leukocyte adhesion cascade, Leukocyte adhesion deficiency (LAD) types I-IV, have been described [2]. In this review, we focus on LAD I (OMIM #116029).

LAD I is caused by mutations in the *ITGB2* gene (21q22.3) encoding CD18, a β2 integrin, which builds heterodimeric cell surface receptors with the four α subunits CD11a, CD11b, CD11c, and CD11d [3]. One heterodimer is LFA-1 (α_L_β_2,_ CD11a/CD18). α_L_β_2_ is expressed on all leukocytes, while α_M_β_2_ (Mac-1, macrophage-1 antigen, CD11b/CD18), α_X_β_2_ (P150,95, CD11c/CD18), and α_D_β_2_ (CD11d/CD18) are mainly expressed on myeloid cells [4]. Since the β and α subunits pair intracellularly, a decreased or mutated β subunit also leads to a decreased expression of the α subunit at the cell surface. All of the mentioned dimers that share the β2 subunit were found to be affected in LAD I [5]. 

The complex CD11a/CD18 is called LFA-1, which is able to bind six ligands to mediate leukocyte arrest and adhesion to the endothelium: ICAM-1, ICAM-2, ICAM-3, ICAM-4, ICAM-5 (intercellular adhesion molecules 1–5), and JAM-A (junctional adhesion molecule A) [6,7,8,9] (Figure 1). β1 integrins such as α_4_β_1_ (VLA-4, very late antigen 4) also participate in leukocyte rolling and firm adhesion. Since VLA-4 is mainly expressed on monocytes and T cells, but not on neutrophils, it is not able to compensate for the loss of β2 integrins in LAD I [10,11].

Patients suffering from LAD I, often characterized by less than 2% CD18 expression on the neutrophils due to bi-allelic mutations in *ITGB2*, present with granulocytosis, delayed detachment of the umbilical cord, inflammatory complications, and recurrent infections of the skin and mucosa without pus formation [12] (Figure 2). Patients with a less severe course of the disease might survive to adulthood and develop periodontitis and bone loss as primarily inflammatory complications of the disease, rather than infections per se [13]. Moutsopoulos et al. [14] found an elevated IL-23 (interleukin-23) and IL-17 (interleukin-17) signature at inflamed sites and treated one adult LAD I patient with the monoclonal anti-IL12/IL-23 antibody ustekinumab, successfully ameliorating his inflammatory symptoms without exacerbating his infections. For severely affected patients, alloHSCT remains the only available curative treatment option. If untreated, survival to the age of two years for patients with severe LAD I was reported to be about 39%, summarized in a review of all published LAD I cases between 1975 and 2017 [15]. A recently introduced lentiviral-mediated gene therapy for patients with severe LAD I by Rocket Pharmaceuticals demonstrated safety and efficacy in phase 1 clinical trials and, thus, may be able to provide an alternative to alloHSCT in the future [16].

LAD II results from mutations in the *SLC35C1* gene encoding a specific GDP-fucose transporter in the Golgi apparatus. Fucosylated oligosaccharides play an essential role in selectin ligand recognition, whereas β2 integrin expression and function are not impaired in LAD II. Neutrophils of affected patients are unable to roll on activated endothelial cells [17,18]. Patients show syndromic features, severe growth retardation, recurrent bacterial infections, and the Bombay blood group [19].

The underlying defects in LAD III are mutations in the *FERMT3* gene encoding Kindlin-3 that regulates the activation of all integrins on immune cells and thrombocytes. By binding the kindlins to the β integrin tails, platelets shift to a high-affinity state [20]. It has been shown that the conformational activation and adhesiveness of LFA-1 on Kindlin-3-null T and B cells is impaired [21]. LAD III is characterized by a Glanzmann-type bleeding syndrome and in some reported cases involved malignant infantile osteopetrosis [22,23]. 

In addition to that, Sorio et al. [24] found a specific monocyte adhesion deficiency caused by gene mutations in the *CFTR* gene of patients with cystic fibrosis (CF) and called it LAD IV. In CF monocytes, the activation of RhoA (Ras homolog gene family member A) and CDC42 (cell division control protein 42) are defective, resulting in an impaired integrin activation [25]. Therapeutic strategies include supportive antimicrobial treatment, local wound management, enzyme replacement for LAD II, CFTR-correcting drugs for LAD IV, and alloHSCT, HSCT being the only available curative treatment option at the moment for LAD I and III. 

In a joint effort with the European Society for Blood and Bone Marrow Transplantation (EBMT), we analyzed data of 84 LAD I and LAD III patients from 33 centers who underwent alloHSCT in the years 2007 to 2017. The 3-year overall survival was 83% for the entire cohort, being superior in patients having a matched sibling donor or a 10/10 matched unrelated donor and being younger than 1 year. Nevertheless, the event-free survival rate in this cohort, i.e., being a survivor without graft failure or graft-versus-host disease (GVHD), was only 58% for LAD I patients and 56% for LAD III patients. Despite high transplant standards, a significant percentage of patients suffered from graft failure or GVHD as well as inflammatory complications during the course of alloHSCT. The cumulative incidence for primary and secondary graft failure at 36 months after HSCT was 17%, while the cumulative incidence for severe acute GVHD grade II-IV was 24% after 100 days. While most of the patients with graft failure could be rescued with a second transplant, acute GVHD was a significant cause of death [26].

Based on our data and findings by Moutsopoulos et al. [14], we postulated that the inflammatory microenvironment caused by LAD may be responsible for the distinct inflammatory complications during alloHSCT. The question of an anti-inflammatory pretreatment remains to be investigated further. With this review, we aim to provide an overview of the intrinsic role of inflammation in LAD I based on the available literature via PubMed searches.

## 2. Structure of LFA-1 and Important Mutations in LAD I

Integrins are the major receptors for mediating cell–cell interactions, adhesion to the extracellular matrix, and activation of many intracellular signaling pathways. In humans, eight β subunits can associate with eighteen α subunits to form a total of twenty-four distinct heterodimeric integrins [27]. Each integrin subunit consists of a long glycosylated extracellular domain, a transmembrane domain, and a short intracellular domain, localized in the cytoplasm [28]. The adhesiveness and signaling function of integrins can be regulated by processes called inside-out and outside-in signaling. Inside-out signaling leads to conformational changes in integrins by the activation of intracellular signaling pathways subsequent to the binding of chemokines and cytokines to their receptors. In this way, the integrin is converted from a folded, inactive, low-affinity or resting state to an extended, high-affinity form [29]. Outside-in signaling refers to integrin activation induced by the binding of extracellular ligands [30].

LFA-1 has three alternative conformational states. The bent, low-affinity state has the headpiece close to the plasma membrane, while the intermediate state, with straightened extracellular leg domains, has a closed headpiece. The high-affinity state has an extended extracellular domain and an opened headpiece allowing interaction with its ligands [31,32] (Figure 3).

As mentioned earlier, LAD I is caused by mutations in the *ITGB2* gene, which encodes the β2 integrin. The *ITGB2* gene is located at 21q22.3 and spans a region of 40 kb which can be divided into 16 exons. Van de Vijver et al. [34] summarized a total of 86 allelic mutations in 123 patients; most of the point mutations were found in the β1-domain, coded for by exons 5–9 of the *ITGB2* gene, which builds the ligand-binding site together with the α-subunit. Other regions for missense mutations in the β2 subunit comprise the last two cysteine-rich repeats, which provide structural stability and are encoded by exon 13 [35]. In a large Indian cohort of LAD I patients, Sanger sequencing of the *ITGB2* gene identified 57 mutations in 105 patients; 54% of these mutations were located in exons 5–9, with most of them being clustered in exons 6 and 7, followed by 32% of mutations in the cysteine-rich repeat region. Most of the mutations were missense (40%) [36] (Figure 4).

Most point mutations are located in the β1 domain (orange) and the cysteine-rich domain (yellow-green) of CD18.

## 3. LFA-1 and Its Interacting Partners

Leukocyte rolling and adhesion on the endothelial layers are mediated by the binding of LFA-1 to its ligands of the immunoglobulin superfamily, ICAM-1, ICAM-2, ICAM-3, ICAM-4, and ICAM-5, as well as JAM-A, which are expressed on the luminal surface of activated endothelial cells during inflammation [6,7,9] (Figure 1). ICAM-1 is normally expressed at low levels and can be upregulated by several proinflammatory cytokines such as TNF-α (tumor necrosis factor alpha) or IL-1β (interleukin 1 beta), whereas ICAM-2 is expressed at stable levels on endothelial cells and does not show induction upon stimulation [39,40]. ICAM-3 is highly expressed in all leukocytes and has been shown to be expressed on the surface of apoptotic leukocytes, which can then be identified and phagocytosed by macrophages [41,42]. ICAM-4, formerly known as the LW (Landsteiner–Wiener) blood group antigen, is a glycoprotein exclusively expressed on red blood cells and erythroid precursor cells, while ICAM-5, also called Telencephalin, is expressed in the central nervous system [43,44]. JAM-A has also been identified to contribute to the transendothelial migration of neutrophils and T cells by binding to LFA-1 [9].

LFA-1 is strongly expressed by T-lymphocytes and is essential for T cell recruitment to inflammatory sites and T cell activation by binding to ICAM-1 on endothelial or antigen-presenting cells to form an immunological synapse [45]. Numerous positive regulators of LFA-1 activation such as Talin, RapL (regulator of adhesion and cell polarity enriched in lymphoid tissues/Nore 1B, Rassf5), ADAP (adhesion and degranulation-promoting adapter protein), SKAP55 (Src kinase-associated phosphoprotein of 55kDa), and MST1 (macrophage stimulating 1) have been reported [46]. Another important effector for integrin activation is Kindlin-3, which is mutated in LAD III and leads to life-threatening infections and bleeding complications [21,47]. LFA-1 on regulatory T cells (Tregs) is essential for their suppressor function. Similar to Tregs from LFA-1-deficient patients, the blocking of LFA-1 on Tregs with anti-CD18 or anti-CD11a antibodies leads to an impaired suppression of mouse T cell activation and proliferation [48]. LFA-1 expressed on cytotoxic T cells mediates the induction of apoptosis of target cells [49]. Aside from VLA-4 and PSGL1 (P-selectin glycoprotein ligand 1), LFA-1 is strongly expressed on memory T cells, and the combination of several adhesion molecules on the surface allows T cells to migrate to different peripheral sites [50]. On NK cells, LFA-1 is involved in activation and lytic synapse formation [51]. 

During B cell synapse formation, it has been shown that the interaction of LFA-1 with its ligand ICAM-1 is able to increase the adhesion capacity of B cells; thus, lower antigen amounts are needed for B cell activation [52]. In an autoimmune mouse model, the highest expression of LFA-1 was found on memory B cells. The use of blocking antibodies against LFA-1 and VLA-4 led to a release of memory B cells from the spleen to the peripheral blood, suggesting a role for integrins in B cell trafficking [53]. Myeloid cells such as monocytes and neutrophils use Mac-1 and LFA-1 for crawling in activated venules. While neutrophils mainly use Mac-1, monocytes switch between LFA-1 and Mac-1 [54]. 

## 4. LFA-I and Neutrophil Function beyond the Antimicrobial Defense 

For many years, neutrophils were only recognized as the main anti-microbial effector cells of the innate immune system, but in the last decade, several immunomodulatory functions have been attributed to them.

As neutrophils are relatively short-lived cells that are generated and released from the bone marrow at a rate of 10^11^ per day, fine regulatory mechanisms are needed to maintain neutrophil homeostasis in the human body [55]. One major promoter of neutrophil production, differentiation, and their release from the bone marrow into the circulation is G-CSF (granulocyte colony-stimulating factor) [56]. In response to several stimuli and cytokines, including LPS (lipopolysaccharide), TNF-α, IFN-γ (interferon-γ), IL-3 (interleukin 3), and GM-CSF (granulocyte-macrophage colony-stimulating factor), G-CSF is produced by monocytes, macrophages, and fibroblasts [57]. Upstream, the cytokine IL-17 (interleukin 17) regulates granulopoiesis by inducing G-CSF and suppressing inhibitors of the leukocyte adhesion cascade such as Del-1 (developmental endothelial locus-1) [58]. When fibroblasts are cultured in the presence of IL-17, the proliferation of CD34+ progenitors and their maturation into neutrophils are maintained [59]. Furthermore, IL-17 is known to induce a series of proinflammatory cytokines and the expression of ICAM-1 [60]. IL-17 itself is regulated by IL-23, which is mainly produced by dendritic cells and macrophages [61].

Similar to patients with LAD I, mice deficient in leukocyte adhesion molecules display neutrophilia [62]. The first explanation to be derived for this condition was the passive accumulation theory, which was based on impaired neutrophil migration out of the blood vessel to peripheral tissues and enhanced neutrophil survival.

Forlow et al. [63] investigated the underlying mechanisms of high neutrophil counts in leukocyte adhesion-deficient mice and generated chimeric mice with different ratios of CD18+/+ and CD18−/− circulating neutrophils. The presence of only 10% CD18+/+ neutrophils was sufficient to prevent neutrophilia in CD18−/− mice, indicating that the intravascular accumulation of poorly adherent neutrophils is not the only cause of the high neutrophil levels. They also found significantly elevated serum and plasma levels of G-CSF and IL-17 in CD18−/− mice corresponding to the levels of neutrophilia in these mice. Another study showed that the lifespan of CD18-deficient neutrophils in the blood circulation and in the bone marrow was not increased compared to wild-type neutrophils [64].

From observations in LFA-1-deficient mice, the concept of a feedback loop called the “neurostat”, which measures and regulates neutrophil numbers, was introduced. According to this model, phagocytosis of apoptotic neutrophils in peripheral tissues downregulates IL-23 secretion from macrophages and dendritic cells, which results in reduced IL-17 and G-CSF expression of Th17 cells. In adhesion molecule-deficient mice, the impaired migration of neutrophils out of the blood and the reduced neutrophil uptake by macrophages leads to high levels of IL-23 and, subsequently, elevated IL-17 and G-CSF [65]. The incubation of CD18−/− splenocytes with rIL-23 was observed to stimulate IL-17 production in a dose-dependent manner. In contrast, when CD18−/− splenocytes were co-cultured with LPS-stimulated dendritic cells and an antibody against IL-23 was added, IL-17 production was inhibited. In addition, the transfer of wild-type bone marrow neutrophils into CD18−/− mice significantly decreased IL-23 expression and the serum IL-17 levels were reduced by 52% [66].

## 5. LFA-I and the IL-12/IL-23 Pathway

The intrinsic role of inflammation in LAD I has been discussed in several human and animal models of the disease.

As mentioned earlier, LAD I is associated with periodontitis and inflammatory bone loss. Whilst oral infections have been linked to impaired neutrophil surveillance in the periodontal tissue for many years, Moutsopoulos et al. [13] recognized the involvement of the IL-23/IL-17 axis in LAD I. They found an excessive production of mainly T cell-derived IL-17 and an elevated expression of cytokines associated with the induction of IL-17, such as IL-1β, IL-6, and IL-23, in the inflammatory lesions of the gingival tissue of LAD I patients and LFA-1 knockout mice. Furthermore, chemokines or cytokines that are involved in granulopoiesis and neutrophil recruitment, such as G-CSF, CXCL2 (C-X-C motif chemokine ligand 2), and CXCL5 (C-X-C motif chemokine ligand 5), were also upregulated.

PCR-based quantification of the bacterial load in the gingival tissue showed no differences between LAD I patients and healthy controls, whereas LFA-1 knockout mice displayed higher periodontal bacterial counts than wild-type controls. In LFA-1 knockout mice being treated with anti-IL17A antibody or anti-IL23p19 antibody, the expression of IL-17 was diminished, and the mice were protected from inflammatory bone loss. They also exhibited a lower bacterial burden, suggesting not only an anti-inflammatory but also an anti-microbial effect of the antibodies. Further exploration of the subgingival microbiome has revealed differences between healthy and LAD I patients. The subgingival LAD I plaques and their products, such as lipopolysaccharide (LPS), are able to trigger IL-23 responses in vitro and in vivo [67]. The authors reported a 19-year-old male with LAD I who suffered from severe periodontitis as well as large sacral lesions and recurrent infections. Staining and flow cytometric analysis of the inflamed sites revealed dense infiltrates of IL-17-producing cells within the lesions. Targeting IL-17 via a blockade of the IL-23/IL-17 axis with the IL-12/IL-23 antibody ustekinumab led to the dramatic improvement of the oral inflammation and the sacral wound [14].

Chronic colitis, resembling Crohn’s disease, with extensive inflammation and ulceration of the terminal ileum due to impaired neutrophil function, has been reported to be another inflammatory complication in LAD I patients [68,69]. In an observational study, the use of ustekinumab was found to be efficacious and safe in children with inflammatory bowel disease [70].

## 6. Druggable Targets in LAD I

Moutsopoulos et al. [14] could show that the overexpression of the proinflammatory cytokines IL-23 and IL-17 in LAD I-associated periodontitis and blocking of the p40 subunit of IL-23/IL-12 via ustekinumab led to a resolution of the inflammatory lesions and diminished IL-17 levels. In a recent study, the group treated CD18−/− mice with agonists of the transcription factors LXRα/LXRβ (liver X receptors α/β) and PPAR β/δ (peroxisome proliferator-activated receptors β/δ) which are known to promote the resolution of inflammation by the regulation of efferocytosis and neutrophil homeostasis [71,72]. Indeed, pharmacological induction of these receptors resulted in decreased expression of IL-23 and IL-17 and improved bone levels in CD18−/− mice [73].

Marsili et al. [74] report the case of a twelve-year-old girl with LAD I and Crohn’s-like colitis and arthritis who was treated with the monoclonal anti-TNF-α antibody infliximab due to a poor response to conventional therapy with prednisolone and mesalamine. After 30 months of treatment, the inflammatory symptoms improved and no relevant side effects occurred (Table 1), (Figure 5).

Putting aside the inflammatory aspects of LAD I, the therapeutic targeting of integrins, due to their involvement in leukocyte recruitment in inflammatory diseases, has been subject to several studies. Efalizumab, a monoclonal antibody against CD11a which blocks the interaction of LFA-1 and its ligand, ICAM-1, has been shown to ameliorate psoriatic skin lesions, but it was not effective in the treatment of psoriatic arthritis [76]. In 2009, the antibody was withdrawn from the market because of an increased risk of John Cunningham (JC) polyomavirus reactivation and the development of progressive multifocal leukoencephalopathy (PML) [77]. Natalizumab, an antibody that targets the α4-integrin subunit, has been approved for the treatment of relapsing multiple sclerosis and Crohn’s disease; however, this has also shown adverse events such as JC (human polyomavirus 2) virus reactivation, indicating a multifaceted role for integrins in the immune system [78,79].

## 7. Conclusions and Perspectives

LFA-1 is critical for mediating leukocyte adhesion, trafficking, and forming a synapse between different immune cells. Patients suffering from LAD I have dysfunctional or absent integrins, which not only lead to infectious problems caused by impaired neutrophil recruitment but also trigger inflammatory complications, even during the procedure of an alloHSCT. With a better understanding of the underlying cellular mechanisms, the identification of new therapeutic targets may be possible. Detailed bioinformatic analyses such as proteomics and deep RNA sequencing data are not yet available for LAD I; nevertheless, their advent should help identify new and better targets. 

## Figures and Tables

**Figure 1 ijms-23-03578-f001:**
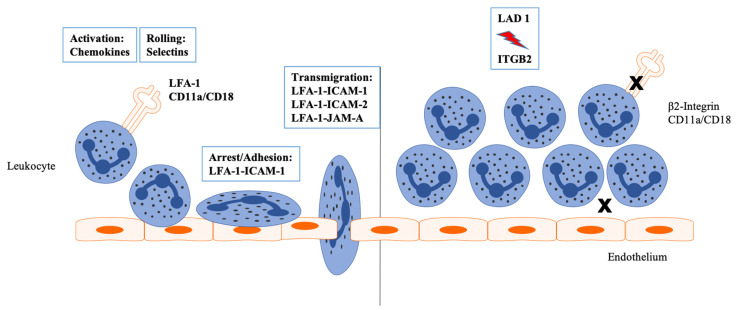
The leukocyte adhesion cascade. Inflammation leads to an activation of the endothelium by endogenous and exogenous stimuli. Selectins, expressed on the surface of activated endothelial cells, mediate rolling of the leukocytes on the luminal surface of the blood vessel. Interactions of the β2-integrin LFA-1 (lymphocyte function-associated antigen 1) and ICAM-1 and ICAM-2 (intercellular adhesion molecule 1 and 2) enable leukocytes to adhere to the inflamed endothelium. Paracellular and transcellular migration of the leukocytes through the vessel walls is then triggered by ligation of JAM-A (junctional adhesion molecule A), ICAM-1, and ICAM-2. In LAD I, absent or defective expression of β2-integrins results in an impaired transmigration of leukocytes.

**Figure 2 ijms-23-03578-f002:**
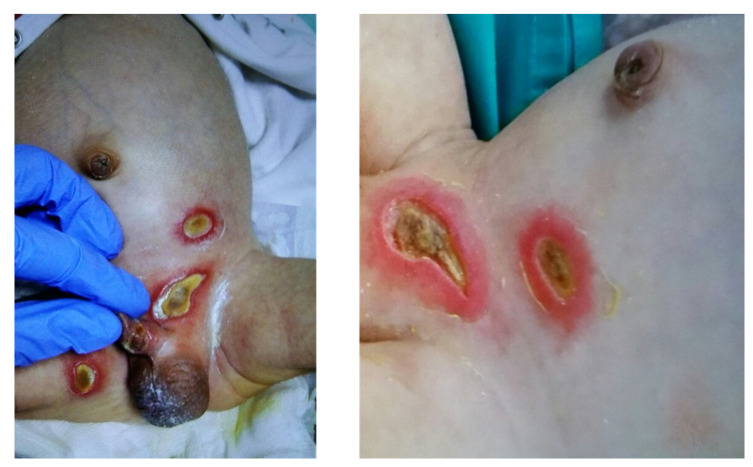
Clinical signs. Lesions of the skin and mucosa without pus formation in a patient with severe LAD I.

**Figure 3 ijms-23-03578-f003:**
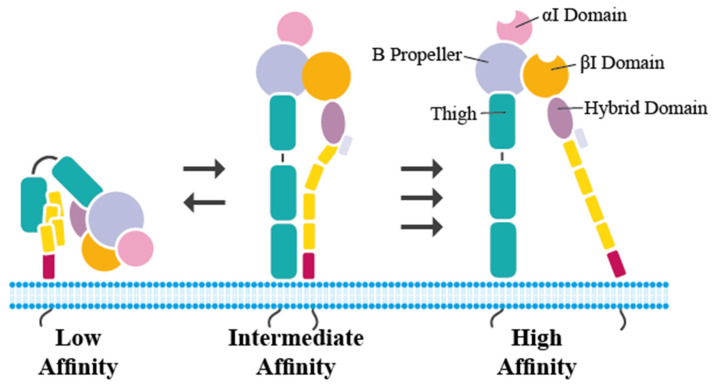
Conformational states of LFA-1 (adapted from Walling et al. [33]). LFA-1 exists in three different conformational states: The low-affinity state (bent-closed) with the headpiece close to the plasma membrane, the intermediate affinity state (extended-closed) with straightened legs, and a closed headpiece and the high-affinity state (extended-open) with extended legs and an open headpiece.

**Figure 4 ijms-23-03578-f004:**
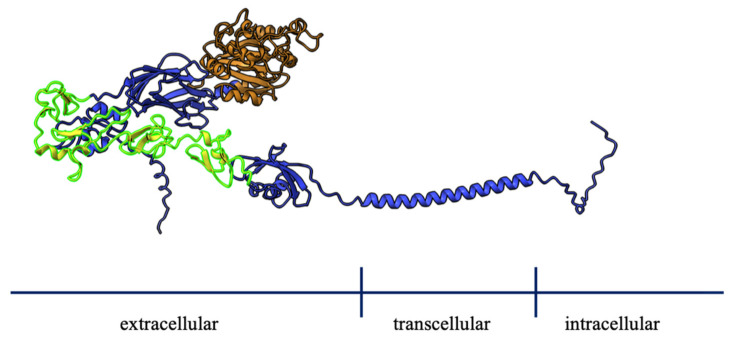
Protein structure of CD18 (graphics performed with AlphaFold [37] and ChimeraX-1.3 [38]).

**Figure 5 ijms-23-03578-f005:**
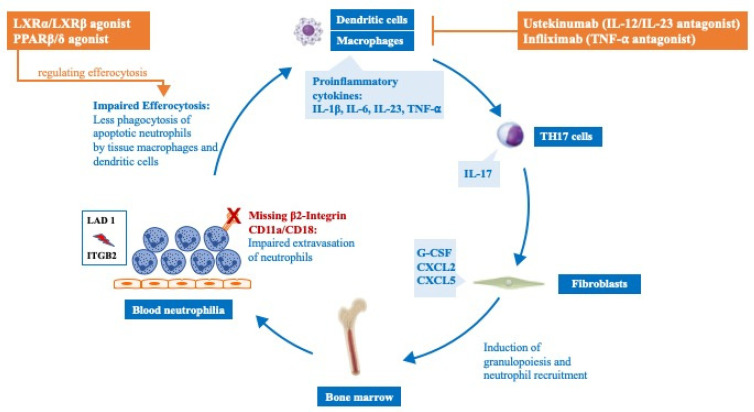
Inflammatory aspects of LAD I (adapted from Hajishengallis et al. [75]) Impaired migration of CD18-deficient neutrophils to the tissue leads to a disruption of neutrophil homeostasis in LAD I. This triggers the overproduction of IL-23 by macrophages and dendritic cells and downstream IL-17 produced by Th17-cells. IL-17 induces the production of G-CSF by fibroblasts, which results in excessive production and release of neutrophils from the bone marrow to the peripheral blood. The use of pharmacological inducers of neutrophil homeostasis regulators (LXRα/LXRβ and PPAR β/δ) and IL12/IL23 via Ustekinumab succeeded to cut off this inflammatory cascade.

**Table 1 ijms-23-03578-t001:** Druggable targets in LAD-1.

Drug	Target	Mechanism	Results
Ustekinumab	IL-23/IL-12	Binds p40 subunit and blocks IL12/IL23 receptor interaction	Decreased IL-17 levels, resolution of inflammatory lesions in an adult patient
LXRα/LXRβ agonist	LXRα/LXRβ	Activation of LXR promotes clearance of apoptotic cells in macrophages	Decreased IL-23 and IL-17, improved bone levels in CD18−/− mice (*71*)
PPARβ/δ agonist	PPARβ/δ	Activation of PPAR leads to regulation of efferocytosis	Decreased IL-23 and IL-17, improved bone levels in CD18−/− mice (*72*)
Infliximab	TNF-α	Binds and neutralizes TNF-α	Improvement of Crohn’s-like colitis and arthritis in LAD 1 patient (*74*)

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
