# Peer review of "Understanding the Role of LFA-1 in Leukocyte Adhesion Deficiency Type I (LAD I): Moving towards Inflammation?"

_ijms, 2022, doi:10.3390/ijms23073578_

Round 1

Reviewer 1 Report

In this study by Fekadu and colleagues, a role of inflammation in the pathophysiology of leukocyte adhesion deficiency type I (LADI) is outlined. The authors describe LADI as a deficiency of integrin subunit beta 2 and explain that beyond the direct functional defects, there is also a role for interleukin 17- and 23-driven inflammation. The subject is interesting, but the paper as a whole could benefit from a number of clarifications and improvements:

  • line 34: the modern name for JAM-1 is JAM-A. The cited paper is over 20 years old and there might be newer studies on this subject.
  • Figure 1: the authors correctly state that a number of integrins (mac-1, alphax beta2) are also dependent on beta 2, but it is unclear whether defects related to LADI also affect these integrins. This should be mentioned. Why can't alpha4 beta 1 (VLA4) compensate for loss of LFA-1?
  • line 46: abbreviations appear at the wrong place
  • section 2 and section 3: I think that section 2 can be expanded. This for example by combining it with the lines 101-120 of section 3 and by adding a figure with a scheme of the different structures of LFA-1 and of the locations of a number of notable point mutations (or a hotspot of mutations, e.g. the I domain).
  • line 138: perhaps it is worth adding kindlin-3 here with reference to LADIII
  • line 167: please specify where the neutrophils migrate to. Back to the bone marrow? To tissues? Does LFA-1 deficiency also affect lifespan and clearance of neutrophils?
  • line 177 can be much clearer: which phagocytes take up neutrophils and where? Which cells express IL23? "Phagocytes" can be any cell with phagocytic potential, including the neutrophil.
  • The locations of the references to table 1 and figure 3 are not optimal. Figure 3 should contain more mechanistic information (e.g. phagocytosis) and a more extensive legend with explanations (e.g. role of fibroblasts). Table 1 should contain the references to the original studies.
  • the German symbol "Eszett" appears to used to write Greek beta in this manuscript. These are in fact different symbols (see e.g. line 224, LXRβ vs PPARß)

Author Response

 Reviewer 1:

  1. line 34: the modern name for JAM-1 is JAM-A. The cited paper is over 20 years old and there might be newer studies on this subject.

Ad 1. Thank you for this comment. We changed the name from JAM-1 to JAM-A throughout the text and added newer studies (line 40-41).

  1. Figure 1: the authors correctly state that a number of integrins (mac-1, alphax beta2) are also dependent on beta 2, but it is unclear whether defects related to LADI also affect these integrins. This should be mentioned. Why can't alpha4 beta 1 (VLA4) compensate for loss of LFA-1?

Ad 2. Thank you for this comment. aMβ2, aXβ2 and aDβ2 share the β2 subunit und are therefore all affected in LAD I (line 31-37).

Since VLA-4 is mainly expressed on monocytes and T cells, but not on neutrophils, it is not able to compensate the loss of β2 integrins in LAD-I (lines 42-44).

  1. line 46: abbreviations appear at the wrong place

Ad 3. Thank you for this comment. We inserted the abbreviations in the legend of figure 1 (lines 48-55).

  1. section 2 and section 3: I think that section 2 can be expanded. This for example by combining it with the lines 101-120 of section 3 and by adding a figure with a scheme of the different structures of LFA-1 and of the locations of a number of notable point mutations (or a hotspot of mutations, e.g. the I domain).

Ad 4. Thank you for this comment. We expanded section 2 by combining it with section 3 and inserted a figure of the 3 conformational states of LFA-1 (figure 3) and one figure that shows the most common mutations in the CD-18 domains (figure 4) (lines 118-162).

  1. line 138: perhaps it is worth adding kindlin-3 here with reference to LADIII

Ad 5. Thank you for this comment.  We added Kindlin-3, which is affected in LAD III, to activators of integrin activation (line 185-187).

  1. line 167: please specify where the neutrophils migrate to. Back to the bone marrow? To tissues? Does LFA-1 deficiency also affect lifespan and clearance of neutrophils?

Ad 6. Thank you for this comment. In LAD I neutropihls fail to  migrate from the blood vessel to the tissue (lines 226-227). A study could show that the lifespan of CD18 deficient neutrophils in the peripheral blood and in the bone marrow was not increased compared to wildtype neutrophils (lines 234-236).

  1. line 177 can be much clearer: which phagocytes take up neutrophils and where? Which cells express IL23? "Phagocytes" can be any cell with phagocytic potential, including the neutrophil.

Ad 7. Thank you for this comment. We edited this paragraph. The phagocytosis of apoptotic neutrophils in peripheral tissues downregulates IL-23 secretion from macrophages and dendritic cells which results in reduced IL-17 and G-CSF expression of Th17 cells (lines 239-241).

  1. The locations of the references to table 1 and figure 3 are not optimal. Figure 3 should contain more mechanistic information (e.g. phagocytosis) and a more extensive legend with explanations (e.g. role of fibroblasts). Table 1 should contain the references to the original studies.

Ad 8. Thank you for this comment. We changed the position of the references to table 1 and the figure (now figure 5). The Figure and the associated legend were complemented with additional mechanistic information and explanations (lines 298-306). References were added to table 1.

  1. the German symbol "Eszett" appears to used to write Greek beta in this manuscript. These are in fact different symbols (see e.g. line 224, LXRβ vs PPARß)

Ad 9. Thank you for this comment. The symbol Greek beta is now used throughout the text.

Reviewer 2 Report

Authors review the role of inflammation in Leukocyte adhesion deficiency type I (LADI) patients. It raised the following questions and suggestions:  

  1. A brief introduction of LAD types I-IV and their relevance with LFA-1 is needed.
  2. Authors reviewed the relationship of LFA-1 and neutrophil function and briefly mentioned its function in T lymphocytes. To obtain an overview, it would be helpful to review the relationship of LFA-1 and other subsets of leukocytes and lymphocytes, which play extensive and fundamental roles in inflammation of LADI patients. 

Author Response

  1. A brief introduction of LAD types I-IV and their relevance with LFA-1 is needed.

Ad1. Thank you for this comment. We added an introduction of LAD types II-IV to section 1 (lines 77-97).

  1. Authors reviewed the relationship of LFA-1 and neutrophil function and briefly mentioned its function in T lymphocytes. To obtain an overview, it would be helpful to review the relationship of LFA-1 and other subsets of leukocytes and lymphocytes, which play extensive and fundamental roles in inflammation of LADI patients.

Ad2. Thank you for this comment. In section 3 we added other subsets of leukocytes and their interaction with LFA-1 (lines 190-203).

Round 2

Reviewer 1 Report

The manuscript was sufficiently revised. Be sure to check the correct layout of figure 4.

Author Response

 Reviewer 1:

  1. Be sure to check the correct layout of figure 4.

Ad 1. Thank you for this comment. We changed the design and the layout for figure 4. References were added.

Reviewer 2 Report

Authors have addressed the main suggestions adequately and improved the manuscript significantly. 

Author Response

Thank you for your time and effort.